# Moving north: Warmer waters expand populations of deep-water cartilaginous fishes into Arctic waters

Romaric Jac [1,2,3*], Jon Albretsen [1], Hannes Höffle[1], Robert J. Lennox[4,5], Arved Staby[1], Fabian Zimmermann [1], Claudia Junge[1*]

**1** Institute of Marine Research (IMR), Bergen, Norway, **2** Institut des sciences de la mer de Rimouski, Université du Québec à Rimouski, Rimouski, Quebec, Canada, **3** UMR DECOD (Ecosystem Dynamics and Sustainability), L'Institut Agro, Ifremer, INRAE, Rennes, France, **4** Department of Biology, Dalhousie University, Halifax, Nova Scotia, Canada, **5** Ocean Tracking Network, Department of Biology, Dalhousie University, Halifax, Nova Scotia, Canada

* romaric.jac@uqar.ca (RJ); claudia.junge@hi.no (CJ)

## Abstract

Continental shelf and deep ocean ecosystems are increasingly exposed to anthropogenic pressures including commercial fishing and climate change related environmental stressors. Among the most vulnerable taxa are chondrichthyans due to their life histories with low reproductive output and therefore lower rebound potentials. In temperate regions, many chondrichthyan species are expected to undergo poleward distributional shifts in response to ocean warming. However, the extent and drivers of these shifts remain poorly understood, particularly in deep-water environments. This study aims to assess long-term trends in distribution and abundance for three cartilaginous fish species in the Northeast Atlantic Ocean: the rabbitfish (*Chimaera monstrosa*), the velvet-belly lanternshark (*Etmopterus spinax*), and the blackmouth catshark (*Galeus melastomus*). Focusing on the northern fringe of their distribution, 26 years (1995–2020) of standardised data from Norwegian scientific bottom trawl surveys were analysed using generalized additive models and GIS-based spatial mapping. The results indicate that all three species have undergone significant northward shifts in abundance over the past two decades, although the magnitude and rate varied among species. Several of their prey species exhibited similar latitudinal shifts, suggesting a potential trophic linkage in response to changing thermal regimes. These findings support the hypothesis that warming waters in northern latitudes are driving the poleward redistribution of deep-water chondrichthyans as they seek to remain within their thermal preference ranges. Understanding these spatial responses is critical for informing conservation strategies and future fisheries management in rapidly changing high-latitude marine ecosystems.

**Data availability statement:** The Norwegian coastal survey data are published with the following DOI: 10.21335/NMDC-749719729 and can be accessed through the Norwegian Marine Data Centre (NMDC), here: https://doi.org/10.21335/NMDC-749719729. CTD-measurements from the Norwegian coastal locations off Bud, Eggum and Ingøy are available online from http://www.imr.no/forskning/forskningsdata/stasjoner/index.html in addition to other locations. Temperature and salinity data are available from the global ocean model: Copernicus Global 1/12° Oceanic and Sea Ice GLORYS12 (Copernicus Marine Environment Monitoring Service; https://marine.copernicus.eu). Shapefiles can be found in www.geoboundaries.org and https://www.naturalearthdata.com/.

**Funding:** The author(s) received no specific funding for this work.

**Competing interests:** The authors have declared that no competing interests exist.

## Introduction

Climate change is the main driver of contemporary ecological change [1] and is disrupting biological communities worldwide. Ocean warming directly and indirectly affects species distribution, growth, and mortality, with northern waters being most impacted by global warming [2]. In these waters, climate change is linked to 'Atlantification', a phenomenon closely related to the progression of warm Atlantic water and associated temperature anomalies into polar latitudes [3]. Predicting climate-driven impacts on biodiversity is challenging because biological responses are highly variable [4]. Nonetheless, evidence shows that climate change can alter community diversity and shift the abundance and distribution of mobile marine species (e.g., [5–13]). The magnitude of these impacts depends on the sensitivity (e.g., thermal tolerance), dispersal capacity (e.g., migratory *versus* non-migratory), and ability of the species to exploit new resources (generalist *versus* specialist feeder) and habitats while interacting with resident predators and competitors in newly colonized area.

In the Northeast Atlantic Ocean, biomasses of chondrichthyans (sharks, rays, skates, chimaeras), have declined markedly in recent decades [14,15], with unknown and potentially disruptive ecological effects [16]. Climate-driven shifts in elasmobranch communities have already been documented in the North Sea, where several species exhibit changes in habitat suitability [7]. At the same time, growing recognition of the ecological importance of deep-water chondrichthyans [17,18] has prompted fishery managers and authorities to introduce measures in the Northeast Atlantic, such as total allowable catch reductions and recently direct fishing prohibitions, size, depth and gear restrictions, and partial habitat closures [e.g., [19,20]; NEAFC Rec Rec 08 2024; EU Reg. 2016/2336; EU Reg. 41/2007). Despite this, distributions of most chondrichthyans in the Norwegian Sea (defined by the International Council for the Exploration of the Sea, ICES, www.ices.dk) and the Barents Sea ecoregions (NSE and BSE, respectively) remain poorly defined, while bycatch rates continue to be high [14,21–24].

Understanding distribution patterns of temperate sharks and other chondrichthyans, particularly in deeper waters, is essential for effective species and ecosystem management [25,26]. Yet research on deep-water taxa lags compared to their shallower living counterparts. Species-specific models were implemented to examine long-term distributional changes in three common deep-water bycatch species across northern European waters: the rabbitfish *Chimaera monstrosa* Linnaeus, 1758, the velvet-belly lanternshark *Etmopterus spinax* (Linnaeus, 1758) and the blackmouth catshark *Galeus melastomus* Rafinesque, 1810. All inhabit broad depth ranges (104–1 600 m) in the Northeast Atlantic [27] but are also encountered on shallower continental shelves. *C. monstrosa* extends furthest north, to ~75° N (Dagit & Hareide, 2015; Guallart et al., 2015) and nearly 73° N off Norway [28]. Trophic interactions further shape their distribution and range shifts: *C. monstrosa* feeds primarily on benthic crustaceans, molluscs, and small demersal fishes [29,30], *E. spinax* consumes mesopelagic fishes, cephalopods, and crustaceans [31–33], and *G. melastomus* feeds opportunistically on small fishes, cephalopods, and crustaceans [32,33]. These

consistent dietary patterns across the Northeast Atlantic suggest that prey availability at regional and broader scales may facilitate or constrain northward movements.

Given the paucity of data for these species [27], this study provides a crucial step towards characterising their biogeography and ecology in the Northeast Atlantic. *Chimaera monstrosa* and *E. spinax* are listed as 'Near Threatened' and *G. melastomus* as 'Least Concern' on the IUCN Red List [34–36], yet all three have recently been designated as Protected, Endangered or Threatened Species (PETS) by ICES in the NSE, and in the BSE for *C. monstrosa*. Existing knowledge of their contemporary distribution and habitat preferences [28,37] is insufficient to forecast responses to global change. This study therefore quantifies the influence of shifting environmental variables on these deep-water chondrichthyans at the fringe of their northern distribution range. Demonstrating changes in their biogeographic patterns across an extensive latitudinal range and multi-decadal timeline will help establish a baseline for long-term monitoring and adaptive management.

## Materials and methods

### Time series survey data

This study uses a 26-year (1995–2020) time series to investigate the impact of environmental variation on species distribution, drawing on an annual bottom trawl survey ("Kysttokt"; ICES-acronym: NOcoast-Aco/BTr-Q4) conducted along the Norwegian coast of the NSE and BSE. The dataset includes 18 years (2003–2020) of the "Kysttokt" data and 8 years (1995–2002) of bycatch records from the preceding combined saithe/coastal cod survey (Fig 1; details in [28]). In this study, the entire dataset—referred to as 'Kysttokt'—consists exclusively of scientific trawl data collected using the same sampling design, gear, and depth range (30–590 m), throughout the time series. The stations were allocated using a systematic stratified design along the entire bathymetric gradient, ensuring coverage across shallow, intermediate, and deep strata. The dataset includes 3,159 hauls, each lasting 30 minutes at an average speed of 3.17 knots. A Campelen 1800 bottom trawl ("Shrimp trawl") with a 20 mm mesh size was used, suitable for quantitative capture of all three species, all of which are ≥ 10 cm at birth. The Norwegian Institute of Marine Research (IMR, Norway; and for the earlier years also Nofima, Norway) conducted these surveys between September and December each year. As a result, the data are representative of the distribution of these species during this period of the year. Most bottom trawl stations were fixed and on average, 122 stations were trawled per year. Despite some nonsignificant changes, the spatiotemporal coverage of hauls can be considered homogeneous, thereby minimising the capture bias (S1-S2 Fig and S1 Table).

As fjord water masses differ environmentally from coastal and offshore waters, stations were classified as either 'Outside' or 'Inside' fjord, using the Norwegian Environment Agency's fjord catalogue (https://www.environmentagency.no/). 'Inside Fjord' stations were removed from the dataset. Abundance (individuals per km²) was standardised using trawl distance and a mean trawl opening width of 25 m, for more detail see [28].

### Environmental and spatial data

Following [28], five environmental and spatial covariates were used to explain species presence: maximum trawl depth, near-bottom temperature, near-bottom salinity, distance from the coast, and latitude. Distance to the coast (in km) was calculated using the Near Neighbour Join tool in QGIS. To obtain reliable temperature and salinity data that best reflect conditions throughout the study area since 1995, monthly average temperature and salinity data at every 10m depths were extracted from a global ocean model: Copernicus Global 1/12° Oceanic and Sea Ice GLORYS12 (Copernicus Marine Environment Monitoring Service; https://marine.copernicus.eu). This three-dimensional model includes sea ice dynamics and has a horizontal resolution of 1/12° (9.25 km at the equator and around 4.5 km at subpolar latitudes) and a vertical grid with a resolution of 1w m near the surface to 450 m below 5000 m depth [38]. Monthly averages of salinity and temperature were then associated with each trawl station location (i.e., average latitude, longitude, and depth).

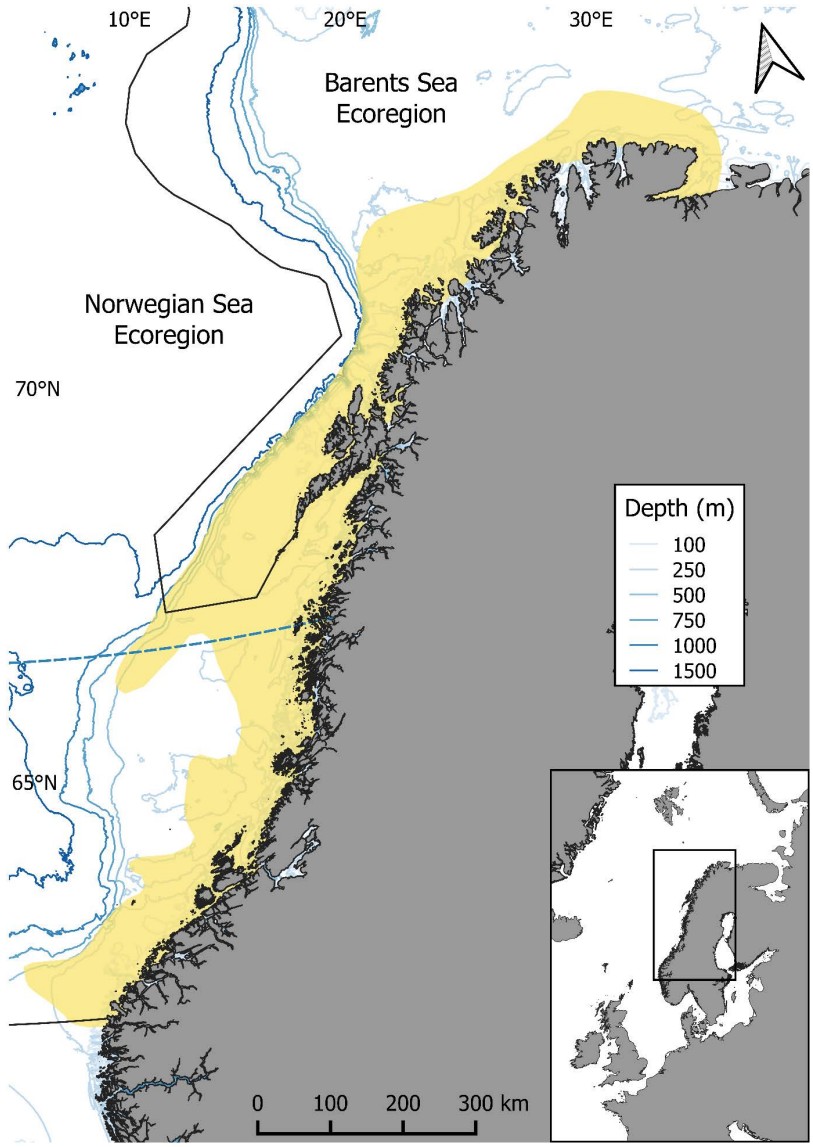

**Fig 1. Norwegian waters showing the depth contour, the area trawled during the Kysttokt survey since 1995 (yellow) and the relevant ICES ecoregions (demarcated by black lines).** Shapefiles used for map construction were obtained from [41].

Although GLORYS has been widely validated, an additional comparison was performed against CTD data collected at three IMR stations —Bud (63°N), Eggum (69°N), and Ingøy (71°N) — for the period 1993 and 2020. Observations at all sites were available down to 300 m depth; however, the corresponding GLORYS grid points were limited to 150 m at Bud and Eggum, and 250 m at Ingøy, due to the model's horizontal resolution and simplified bathymetry. This evaluation confirmed that GLORYS reliably reproduces hydrographical properties in this depth layers and supports the model's suitability for use in this study area, accurately representing the environmental conditions of the studied region. Details are found in the Supplementary text and S3 and S4 Fig. It should also be noted that hydrographic properties used in the statistical analyses at 150, 250, and 350 m were retrieved from model cells located slightly farther offshore in deeper regions.



### Analysis of species habitats

Habitat occupancy (extent of habitat use) and preference (most suitable or favoured habitat) were assessed for each species using the full catch dataset. Even if spatial habitat use may differ among life stages, with neonates, juveniles, subadults, and adults potentially inhabiting different environments, habitat was characterised using five environmental and spatial variables identified in advance as important predictors of demersal chondrichthyan distribution [15,28,39]. Habitat occupancy and preference refer to the sampled population within the depth range of 30–590 m. Habitat preference was determined by identifying the environmental conditions associated with the top 70% of individual captures, representing the most frequently used habitat conditions.

### Changes in abundance and distribution

To evaluate changes in species' abundance and distribution over a 26-year period, three approaches were employed: (1) Generalised Additive Models (GAMs) to analyse spatial distribution; (2) Mapping habitat occupancy at the start (1995) and end (2020) of the time series; (3) Tracking changes in centre of gravity (COG) over time, calculated for depth (30–590 m), latitude (62–72°N), and distance from the coast.

**Model development, selection, and evaluation.** GAMs, an extension of generalised linear models, allow inclusion of smoothed terms (splines) to capture non-linear relationships [40,41]. The models were fitted using the 'mgcv' package in R [24,42], with a negative binomial distribution for catch count data. Predictor smooths were chosen through data exploration and model selection.

Collinearity among predictor variables was tested prior to modelling. Highly correlated variables (Pearson's $r > 0.7$) were excluded [43], as well as variables with variance inflation factor (VIF) values >2.5 [44]. No pairwise correlations (S5 Fig) and multicollinearity among the predictor variables were found.

A forward selection process based on Akaike Information Criterion (AIC; [45]) was used to identify the most parsimonious model. ANOVA tests compared simpler and more complex models at each step. Final models were assessed using AIC, adjusted $R^2$, and explained deviance.

**Mapping the distribution change.** To quantify spatial change over time, habitat occupancy maps were generated for 1995 and 2020 using QGIS, based on shapefiles from [46]. Changes was measured as the proportion of non-overlapping areas between the two years. The systematic stratification of stations across the full depth gradient ensured balanced depth representation, allowing reliable calculation of the COG for each year. The COG for each year was calculated for the three spatial variables by weighting trawl locations by standardised species abundance (approach in [5]). Linear regressions were then used to estimate shifts in species' mean location and depth between 1995 and 2020.

### Temporal changes of potential prey assemblage

To assess possible trophic interactions, binary presence–absence data from the Kysttokt survey were used to evaluate co-occurrence between the three focal chondrichthyans and other species. A species was considered a potential prey if it (1) had a high probability of positive co-occurrence, (2) was morphologically and size suitable as prey and (3) had supporting evidence from other regions. Using the 'cooccur' R package [47], yearly and latitudinal co-occurrence patterns were classified as positive, negative, or random. Co-occurrence frequency was expressed as the percentage of shared hauls relative to the total. Non-random associations may indicate trophic or ecological interactions. Distributional trends of positively associated potential prey species were analysed over time.

## Results

### Time series of bottom water temperatures and salinities

The temperature shows an increasing trend from the 1990s, with an absolute error below one unit (°C/psu) throughout all depths, seasons, and years (S6-S7 Fig). The comprehensive temporal coverage (1995–2020), combined with low error margins, to verifies the use of these data as physical predictors.

Autumn near-bottom water temperatures increased between 1995 and 2016, followed by a decline until 2020 at the northernmost latitude (71°N). No significant temperature differences were found between shallow and deep stations. Waters north of 64°N showed a marked warming trend over most of the study period, with the recent decline around 2020 highlighting the need for further investigation in future studies. The temperature–depth profiles followed expected patterns for the Northeast Atlantic. In contrast, near-bottom salinity remained relatively stable, between 34.1 and 35.7 for 95% of the dataset, with no significant temporal changes detected (S8-S9 Fig).

## Species habitats occupancy and preference

Habitat occupancy and preferences for the three species during the past 26 years are shown in Fig 2 and S2 Table. *Chimaera monstrosa* exhibited the broadest habitat tolerance, occupying a wide range of latitude, and depths (including the maximum depth surveyed), and a broad temperature range (4.1–12.4°C). In contrast, *E. spinax* was found within a more limited geographical extent, associated with cooler waters (up to 10.5 °C). *Galeus melastomus* showed a similarly restricted geographical distribution, with a narrower preferred temperature range (7.2–9.1 °C) and occurring predominantly between 62.7° and 66.4°N. Salinity preferences were narrow across all three species, reflecting the relative homogeneity of near-bottom salinity in the study region rather than specific selection for different salinity conditions.

## Changes in abundance and distribution

The negative binomial abundance model for NSE and BSE identified latitude as the only significant variable across all three species (Table 1). Moreover, for *C. monstrosa*, distance from the coast was also a significant variable (S10 Fig).

Regarding the changes in latitude, over the 26-year period, a northward shift in latitudinal distribution and abundance was evident for all species, with associated changes in habitat occupancy: +26% for *C. monstrosa*, +63% for *E. spinax*, and +55% for *G. melastomus* (Fig 3a). Latitudinal COG time series confirmed these trends. *Chimaera monstrosa* and *E. spinax* each shifted more than +1°N—approximately 120 km northward—since 1995, while *G. melastomus* shifted just under +1°N (Fig 3b). For *C. monstrosa*, the COG crossed the Arctic Circle in 1999 and 2001, and since the 2010s, it has consistently remained above this threshold. The years 2011 and 2017 mark the highest recorded latitudinal values. *Etmopterus spinax* remained mostly south of 66°N, with exceptions in 2013 and 2017. *G. melastomus* showed more stable interannual variation, with a general northward shift of ~70 km over the study period. No significant trends were found in the depth or coastal distance COG for any species.

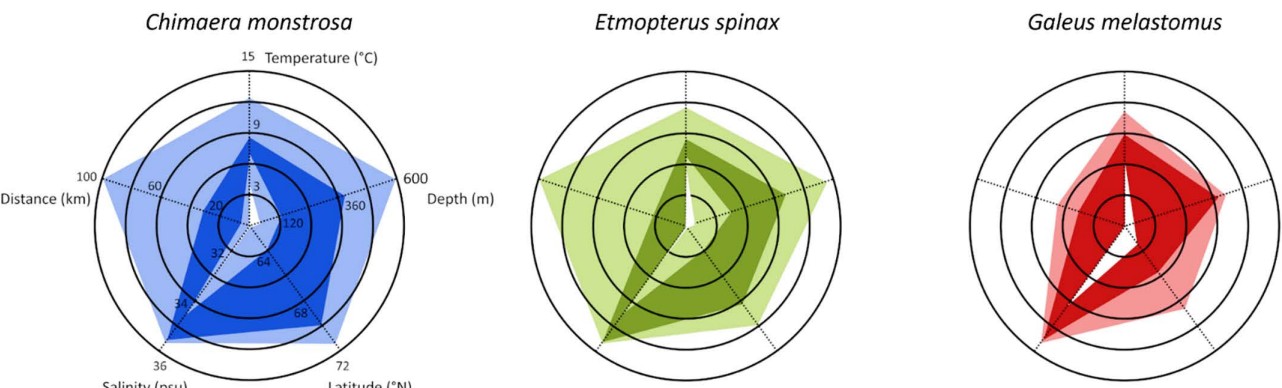

**Fig 2. Radar plot with the environmental variables used to define species habitat occupancy (lighter colour) and habitat preference (70% of individuals; darker colour) over the last 26 years.**



Table 1. Detailed summary of the effect of best-fit and most reduced GAM predictors on the abundance (Negative binomial distribution) for *C. monstrosa*, *E. spinax* and *G. melastomus* in the Norwegian Sea (variables tested: Depth, Latitude and Distance to the coast in function of the year as well as the Year; interactions between variables are indicated by ":"). Models selected by the Adjusted R² (Adj-R²) and the adjusted proportion of deviance explained (Dv exp). "Estimate" refers to the expected change in the response variable for a one-unit change in the predictor, with the associated "t-value" indicating the strength and direction of the relationship and providing evidence against the null hypothesis of a zero-parameter value. Significance value of statistical tests given as ***: p<0.001.

**Model Selection**

| Significant predictors | AIC | Adj R² | Dv exp |
|---|---|---|---|
| *Chimaera monstrosa* | | | |
| NULL | 21835.1 | | |
| s(Year, Lat, k = 15) | 21478.8 | | |
| s(Year, Lat, k = 15) + s(Year, Distance, k = 13) | 21304.5 | 0.460 | 25.0% |
| *Etmopterus spinax* | | | |
| NULL | 11287.7 | | |
| s(Year, Lat, k = 15) | 11025.8 | 0.317 | 24.4% |
| *Galeus melastomus* | | | |
| NULL | 8846.1 | | |
| s(Year, Lat, k = 15) | 7800.3 | 0.546 | 68.3% |

**Parametric coefficients**

| Variables tested | Estimate | Std error | t value | p |
|---|---|---|---|---|
| *Chimaera monstrosa* | | | | |
| Latitude:Year | 4.561 | 0.066 | 68.81 | *** |
| Depth:Year | 5.064 | 0.070 | 72.26 | *** |
| Distance:Year | 5.288 | 0.072 | 73.73 | *** |
| Year | 6.793e-02 | 9.17e-03 | 7.41 | *** |
| *Etmopterus spinax* | | | | |
| Latitude:Year | 3.176 | 0.109 | 29.15 | *** |
| Depth:Year | 4.338 | 0.117 | 37.05 | *** |
| Distance:Year | 4.705 | 0.121 | 39.06 | *** |
| Year | 0.054 | 0.016 | 3.47 | *** |
| *Galeus melastomus* | | | | |
| Latitude:Year | 4.013 | 0.453 | 8.86 | *** |
| Depth:Year | 3.856 | 0.119 | 29.88 | *** |
| Distance:Year | 3.141 | 0.115 | 27.31 | *** |
| Year | 0.012 | 0.015 | 0.78 | |

## Temporal changes of potential prey assemblage

The co-occurrence analysis included 191 species collected during the Kysttokt survey, representing 142 genera. These comprised 86 Osteichthyes, 18 Crustacea, 11 Cephalopoda, 10 Chondrichthyes, 8 Cnidaria, and 3 Gastropoda. Among these species, 24 species co-occurred with at least one of the three focal chondrichthyan species at >1% frequency between 1995 and 2020 (S3 Table).

*Chimaera monstrosa* exhibited the second highest rate of positive co-occurrences among all species, following *Micromesistius poutassou* (blue whiting). However, due to limited sampling of epibenthic fauna and small crustaceans in the Kysttokt survey, which are important components of *C. monstrosa*'s diet, no definitive conclusions could be drawn regarding its prey composition.



**Fig 3. Long-term shifts and preferred habitat of *C. monstrosa*, *E. spinax* and *G. melastomus* in the survey area covered between 1995 and 2020.** These changes are shown with the **(a)** Distribution map of *C. monstrosa*, *E. spinax* and *G. melastomus* for 1995 (orange) and 2020 (purple) showing the habitat occupancy (polygons) and abundance (indiv/km²; dots) as well as the latitudinal centre of gravity in 1995 and 2020 and (b) latitudinal centre of gravity fitted with linear regression. Change through time is written as: slope±SE. Significant value of statistical tests given as *: $p < 0.05$. Shapefiles used for map construction were obtained from [41].

For the two shark species, strong positive co-occurrence patterns were observed with *Trisopterus esmarkii* (Norway pout), *Argentina silus* (greater argentine), *M. poutassou*, *Gadiculus argenteus* (silvery pout), and *Glyptocephalus cynoglossus* (witch flounder). These associations are consistent with previous studies that reported trophic links such as predation by *E. spinax* on *M. poutassou* and *G. argenteus*, and by *G. melastomus* on *Argentina* spp. and *T. esmarkii* [31,48,49]. Further analyses were conducted on these four prey species to assess shifts in their spatial distribution.

Both *G. argenteus* and *T. esmarkii* exhibited significant northward shifts, moving 1.7° and 3° of latitude respectively (Fig 4) during the 26-year period. In contrast, although COG of *A. silus* and *M. poutassou* also moved slightly northward, these shifts were not statically significant. As a result of these shifts, all potential prey species are now predominantly found between 64° and 70°N, with their centres of gravity located largely north of the Arctic Circle.

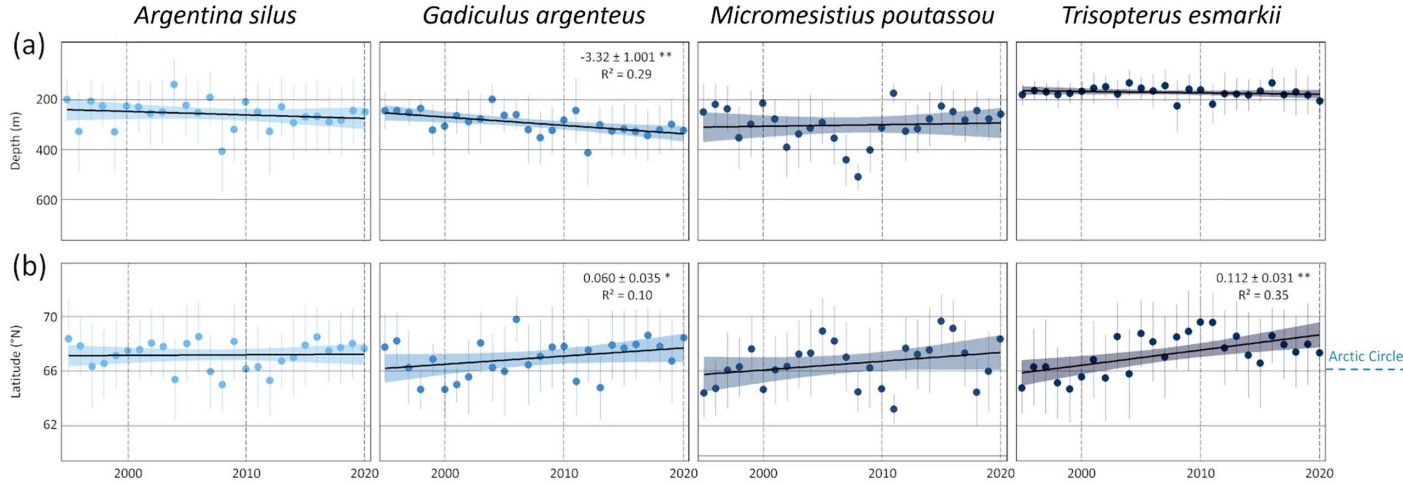

**Fig 4. Long-term abundance shifts in the survey area covered between 1995 and 2020 of the supposed *E. spinax* and *G. melastomus* prey: *A. silus*, *G. argentus*, *M. poutassou* and *T. esmarkii*.** These changes are shown with the change of the (a) depth and (b) latitudinal centre of gravity, fitted with linear regression. Only the slopes with a significative p-value are shown. Significant value of statistical tests given as **: p<0.01, *: p<0.05.

## Discussion

Over the past 26 years, near-bottom water temperatures in the studied area have increased by approximately 1°C (S6 Fig). These results align with previous research highlighting significant warming across the northern European continental shelf, particularly along the Norwegian coast [50], and record-breaking temperatures in the BSE [51]. This warming is primarily attributed to the heat transport from lower latitudes via the Atlantic Meridional Overturning Circulation (AMOC), a major current system regulating North Atlantic climate. Long-term records of the AMOC display large oscillations, with periods of strengthening and weakening during the study period [52,53], which have significantly influenced thermal conditions in northern seas.

### Expanding nordic shark and chimaera populations

These results of this study reveal northward shifts of the COG for *C. monstrosa*, *E. spinax*, and *G. melastomus* by 42, 44 and 24 km per decade, respectively, likely in response to warming in the NSE and BSE. These findings corroborate a growing body of literature documenting poleward shifts in fishes [5–13], consistent with trends observed along the Northwest Atlantic coast [10,54,55]. As chondrichthyans are particularly sensitive to temperature and often migrate to maintain their optimal thermal niche [56], the observed range expansions reflect a broader borealisation of Arctic waters under ongoing Atlantification [3,57], with all three species now established within the NSE and BSE.

Although these shifts describe population-level movements, finer ecological dynamics among life stages are likely masked. Juveniles and adults often occupy distinct depth ranges and habitats [47,48] and may respond differently to environmental change. Changes in the length composition of individuals and the distribution of different length groups across the time series could provide further insight into ontogenetic patterns; however, the present analysis focused on overall population movements. Future studies incorporating stage- or length-specific distributions will be needed to clarify how different life stages contribute to these shifts.

The spatial changes observed were non-linear, punctuated by periods of rapid northward movement coinciding with extreme temperature anomalies. Marine heatwaves (MHWs), defined by [58] as prolonged periods of anomalously warm water, appear to have facilitated temporary and long-term range extensions into more northern and offshore habitats [59].

This process is especially evident in fjord and deep coastal systems, which are thermally sensitive and respond quickly to temperature anomalies [60]. The frequency and intensity of MHWs have risen over the past century [61] and are projected to increase further [62]. Without a slowdown of the AMOC, Norwegian waters could reach near-permanent MHW conditions by 2100 [63], potentially restructuring populations, altering abundance and shifting habitat limits [64] and affecting growth rates [65]. Conversely, a mid-century AMOC collapse, as some scenarios suggest, could amplify climate variability and accelerate ecosystem restructuring [66].

While some individuals may occur at greater depths beyond the range covered by this survey [27], this analysis focused on the main depth interval occupied by *C. monstrosa*, *E. spinax*, and *G. melastomus* in the study area [28], which likely represent their core habitat. Stable abundances in southern survey areas despite northward movements suggest either range expansion or a simultaneous southern contraction outside the sampling domain. Northward migrants may be replaced by individuals originating from regions farther south such as the Skagerrak, Celtic Sea, Faroes, or Iceland where *C. monstrosa* and *E. spinax* also occur. The absence of *Chimaera opalescens* and other chimaeriforms in Norwegian waters, despite their presence farther south [29], hints at a biogeographical barrier that limits northward dispersal. Although no basin-wide study has been conducted, distributional changes for all three species have been reported in the Bay of Biscay [67], supporting large-scale redistribution of deep-water chondrichthyans under warming scenarios.

## Feeding ecology and prey dynamics

Predator distributions are closely linked to ecological interactions such as predation, competition, and facilitation [68]. The ecological roles of *C. monstrosa*, *E. spinax*, and *G. melastomus* have been described as mesopredatory, occupying mid-trophic levels within marine food webs [30–32,69]. Ontogenetic dietary shifts have been documented in these species, involving increases in prey size and changes in prey composition with growth [30,32,70]. However, their functional niches remain poorly understood, particularly for chimaeras, whose trophic roles may vary between mesopredators, scavengers, competitors, or bioturbators depending on context [71,72]. Except for [31], in which 51 individuals of *E. spinax* were collected from southern Norwegian fjords, no extensive studies have examined the feeding behaviour of these species in northern waters, leaving their trophic ecology largely uncertain.

In the present study, trophic drivers were inferred from the co-occurrence of prey species captured in trawl catches. This approach was regarded as a first-order proxy for trophic associations rather than a direct measure of diet. Because trawl catches employed a 20 mm mesh size, only prey large enough to be retained were represented. Smaller or more fragile organisms, such as amphipods, mysids, euphausiids, ostracods, polychaetes, and small cephalopods, may have been underrepresented, despite contributing substantially to the diets of *G. melastomus* and *E. spinax*, particularly during early life stages [32]. Although direct diet composition could not be measured, co-occurrence data provided a valuable preliminary indicator of trophic associations, pending future studies employing stomach-content or stable-isotope analyses (e.g., [31,73,74] to better resolve ontogenetic and habitat-specific dietary variations.

Rising temperatures are expected to reorganise marine food webs [75], altering predator ranges [76,77]. Two of the four main prey species identified, *G. argenteus* and *T. esmarkii*, have shifted significantly northward. *Micromesistius poutassou* has also expanded into the northern and eastern Barents Sea, benefiting from warmer water and reduced sea ice that enhance primary production [78]. Although this analysis focused on prey availability through co-occurrence, substrate type might influence prey habitat. Because benthoscape data were coarse, substrate was not included in the models, but future studies should incorporate finer-scale habitat information and a broader assessment of prey availability. Most prey now reside above the Arctic Circle, resulting in limited overlap with the range of *G. melastomus*. Continued divergence could deprive this shark of key prey resources. Both shark species also feed on crustaceans: *E. spinax* targets *Meganyctiphanes norvegica* [31] and *G. melastomus* targets *Pandalus borealis* [48]. These crustaceans remain widespread in the BSE [79,80], and a dietary switch toward crustaceans may buffer sharks against fish prey loss, as observed in other warming-affected species [81,82]. However, increased reliance on *P. borealis* may elevate bycatch risk, as deep-water

shrimp fisheries capture many chondrichthyans [83]. For *C. monstrosa*, a highly specialized diet, often dominated by *Monodaeus couchii* [69] could limit resilience to climate change [84,85].

## Study limitation and broader implication

This study focused primarily on the influence of warming on species distribution and abundance, yet a broader suite of climate-related stressors may also shape ecological dynamics in northern marine ecosystems. Factors such as sea ice decline, changes in light availability, nutrient fluxes, ocean pollution, acidification, deoxygenation, and shifting current patterns can all drive structural and functional changes in marine communities [86,87]. Additionally, oceanographic variation across basins may mediate species' responses to climate change in distinct ways [84]. Altered hydrographic conditions can also lead to the relocation of spawning grounds and shift fishing activities northward or into deeper waters [88], though such dynamics were beyond the scope of this study. Another limitation lies in the temporal coverage of the dataset: surveys were conducted only between September and December and span a period from 1995 to 2020. While nearly three decades of observations represent a valuable long-term record, this timeframe may still be insufficient to fully capture multi-decadal climatic influences on population dynamics. Moreover, restricting observations to autumn may overlook potential intra-annual shifts in distribution and habitat use. Expanding sampling to additional seasons and extending the time series would clarify whether the documented patterns persist year-round or vary seasonally.

Given the forecasted intensification of Atlantification and extreme climatic events, increased research focus is warranted toward deep-water species at the edges of their distribution ranges. These poorly understood species may serve as important indicators of ecological change. Range shifts are already reshaping species assemblages, with potential ripple effects for ecosystem functioning and human livelihoods [89]. Monitoring and managing new species arriving in Arctic and sub-Arctic waters will become increasingly complex, requiring consideration of both expanding range fronts and contractions at trailing edges [90]. In this study, all three examined cartilaginous species are expanding their ranges and adjusting their abundance to remain within favourable thermal conditions in the NSE and BSE. If current trends persist, further expansion into higher latitudes, such as the White Sea, is plausible. Sustainable management of these transboundary species, particularly in the Barents Sea shared by Norway and Russia, will require strengthened international collaboration, joint monitoring frameworks, and open exchange of data across national boundaries.

## Supporting information

**S1 Fig. Distribution of trawl stations from the Kysttokt survey from 1995 to 2020. Made with Natural Earth on R studio, free vector and raster map data:** https://www.naturalearthdata.com/.
(DOCX)

**S2 Fig. Temporal variation in the (a) latitude, (b) depth, and (c) longitude of Kysttokt survey hauls from 1995 to 2020.**
(DOCX)

**S3 Fig. Time series of the temperature difference between the GLORYS-model and CTD-measurements between 1993 and 2020 at the IMR stations (a) Bud (63°N), (b) Eggum (68°N) and (c) Ingøy (71°N).** Temperature difference above 0 °C means that the model is warmer than the observations.
(DOCX)

**S4 Fig. Time series of the salinity difference between the GLORYS-model and CTD-measurements between 1993 and 2020 at the IMR stations (a) Bud (63°N), (b) Eggum (68°N) and (c) Ingøy (71°N).** Salinity difference above 0 means that the model is more saline than the observations.
(DOCX)



**S5 Fig. Correlation matrix plot (Kendal correlation) with significance levels between Latitude, Depth and Distance from the coast. Statistical significance levels are given as \*\*\*: $p < 0.001$, \*\*: $p < 0.01$, \*: $p < 0.05$.**
(DOCX)

**S6 Fig. Time series of autumn (Sep-Nov) mean temperatures at three locations offshore the Norwegian coast (63, 69 and 71°N) provided by the GLORYS-model.** (a) shows each year's temperatures as dot, with lowpass-filtered time series using 5-year cut-off period as solid lines and the linear trend line as dashed lines. The colours separate the three depths used. Only the linear slopes with a significant p-value are shown. Change through time is written as: slope±SE. Statistical significance levels are given as \*\*\*: $p < 0.001$, \*\*: $p < 0.01$, \*: $p < 0.05$. (b) show autumn temperature with depth for all years (dots) where the line denotes the 1995–2020 average.
(DOCX)

**S7 Fig. Time series of entire year mean temperatures at three locations on the Norwegian coast (63, 69 and 71°N) provided by the GLORYS-model.** (a) shows each year's temperatures as dot, lowpass-filtered time series using 5-year cut-off period as solid lines and the linear trend line as dashed lines. The colours separate the three depths used, which is 150, 250 and 350 m. Only the slopes with a significant p-value are shown. Change through time is written as: slope±SE. Statistical significance levels are given as \*\*\*: $p < 0.001$, \*\*: $p < 0.01$, \*: $p < 0.05$. (b) shows entire year temperature with depth for all years (dots) where the line denotes the 1995–2020 average.
(DOCX)

**S8 Fig. Time series of autumn (Sep-Nov) mean salinities at three locations on the Norwegian coast (63, 69 and 71°N) provided by the GLORYS-model.** (a) shows each year's salinities as dot, lowpass-filtered time series using 5-year cut-off period as solid lines and the linear trend line as dashed lines. The colours separate the three depths used, which is 150, 250 and 350 m. No slopes have a significant p-value. (b) shows autumn salinity with depth for all years (dots) where the line denotes the 1995–2020 average.
(DOCX)

**S9 Fig. Time series of entire year mean salinities at three locations on the Norwegian coast (63, 69 and 71°N) provided by the GLORYS-model.** (a) shows each year's salinities as dot, lowpass-filtered time series using 5-year cut-off period as solid lines and the linear trend line as dashed lines. The colours separate the three depths used, which is 150, 250 and 350 m. No slopes have a significant p-value. (b) shows entire year salinity with depth for all years (dots) where the line denotes the 1995–2020 average.
(DOCX)

**S10 Fig. Long-term shifts of the centre of gravity from 1995 to 2020 of the (top) depth and (bottom) distance to the coast for *Chimaera monstrosa* (blue), *Etmopterus spinax* (green) and *Galeus melastomus* (red).** Change through time is written as: slope±SE. Significant value of statistical tests given as \*: $p < 0.05$.
(DOCX)

**S1 Table. Summary of trawl stations from the Kysttokt survey from 1995 to 2020, indicating the number of trawl stations depth and latitudinal ranges, as well as species occurrence.**
(DOCX)

**S2 Table. Habitat occupancy and habitat preference (70% of individuals) of each species over the last 26 years.**
(DOCX)

**S3 Table. Most co-occurring species and percentage of co-occurrence (min. 1%) during the Kysttokt cruise between 1995 and 2020 with at least one of the species studied: *Chimaera monstrosa, Etmopterus spinax* or**

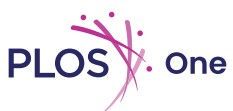

*Galeus melastomus*. The green number represent the number of positive co-occurrence while the red corresponds to the negative ones. The percentage of co-occurrence was calculated from the genus. The percentages calculated correspond to the frequency of observation of these species when *C. monstrosa*, *E. spinax* and *G. melastomus* were caught at a station.
(DOCX)

## Acknowledgments

We would like to thank cruise leaders, research scientists and technicians at the Institute of Marine Research (IMR) in Norway for providing the survey data, Caroline A. Tranang for help with the data preparation, as well as Laurène Merillet and Benjamin Planque for ideas for data analysis and for constructive discussions, and the research group on Deepwater fish, shark, and shellfish (formerly: Deepwater and cartilaginous fishes) for their support and helpful discussions.

## Author contributions

**Conceptualization:** Romaric Jac, Hannes Höffle, Robert J. Lennox, Arved Staby, Fabian Zimmermann, Claudia Junge.

**Data curation:** Romaric Jac, Jon Albretsen.

**Formal analysis:** Jon Albretsen.

**Investigation:** Romaric Jac, Claudia Junge.

**Methodology:** Romaric Jac, Jon Albretsen, Hannes Höffle, Robert J. Lennox, Arved Staby, Fabian Zimmermann, Claudia Junge.

**Supervision:** Claudia Junge.

**Validation:** Jon Albretsen, Claudia Junge.

**Visualization:** Romaric Jac, Jon Albretsen.

**Writing – original draft:** Romaric Jac, Claudia Junge.

**Writing – review & editing:** Romaric Jac, Jon Albretsen, Hannes Höffle, Robert J. Lennox, Arved Staby, Fabian Zimmermann, Claudia Junge.

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
