## [Decision Letter · Decision Letter 0]

3 Oct 2025

Dear Dr. Jac,

We look forward to receiving your revised manuscript.

Kind regards,

Claudio D'Iglio, Ph.D.

Academic Editor

PLOS ONE

Journal Requirements:

2. Thank you for stating the following in your manuscript:

[IMR provided funding through the Norwegian Sea programme for RJ. IMR conducted the scientific surveys to collect the data used here, and financed JA, HH, AS, FZ and CJ.]

[The author(s) received no specific funding for this work.]

5. We note that Figures 1, 3, and S1 in your submission contain map images which may be copyrighted. All PLOS content is published under the Creative Commons Attribution License (CC BY 4.0), which means that the manuscript, images, and Supporting Information files will be freely available online, and any third party is permitted to access, download, copy, distribute, and use these materials in any way, even commercially, with proper attribution. For these reasons, we cannot publish previously copyrighted maps or satellite images created using proprietary data, such as Google software (Google Maps, Street View, and Earth). For more information, see our copyright guidelines: http://journals.plos.org/plosone/s/licenses-and-copyright.

1. You may seek permission from the original copyright holder of Figures 1, 3, and S1 to publish the content specifically under the CC BY 4.0 license.

6. Please include your tables as part of your main manuscript and remove the individual files. Please note that supplementary tables should remain as separate "supporting information" files.

Reviewers' comments:

Reviewer's Responses to Questions

**Comments to the Author**

1. Is the manuscript technically sound, and do the data support the conclusions?

Reviewer #1: Partly

Reviewer #2: Partly

2. Has the statistical analysis been performed appropriately and rigorously?

Reviewer #1: Yes

Reviewer #2: Yes

3. Have the authors made all data underlying the findings in their manuscript fully available?

Reviewer #1: Yes

Reviewer #2: No

4. Is the manuscript presented in an intelligible fashion and written in standard English?

Reviewer #1: Yes

Reviewer #2: Yes

Reviewer #1: The study is of high scientific interest. It is well organized and the results are clearly presented. With regard to the methods, the experimental sampling design should be explained, indicating which statistical approach was used to allocate the stations within the investigated depth range.

The main limitation of this study is that it demonstrates the northward shift of the most superficial fraction of the population of the three species considered. In fact, although the study deals with three deep-sea species, the data sampling partially covers the depth range inhabited by these species. Therefore, some measures of distribution and abundance, such as COG, habitat occupancy and preference, relate to the sampled population in the investigated depth range. It is very likely that they don’t represent the population distributed over the whole depth gradient.

I agree that the authors included a section on the limitations of the study. The aforementioned issues regarding the bathymetric range explored also raise the question of whether populations have moved even deeper than estimated.

Details

Introduction

Line 53. “…ability of the species to exploit new resources…” in presence of predators and competitors in areas of new location.

Materials and Methods

Lines 92-94. Were the data from the two different periods collected using the same sampling design, with the same tool, and within the same depth range?

Line 96. 1800 ma?

Line 103. A bias in the catches is also expected because the investigated depth range does not adequately cover the depth range in which the species are distributed, which is well greater than 1000 m, and 2000 m for E. spinax. Indeed, the species occurrence percentage are low, mostly for G. melastomus and E. spinax. Furthermore, sampling does not allow for the observation of any seasonal variations in the distribution of species throughout the year.

Lines 104-107. The bathymetric range investigated should be also reported in the text. How were the hauls allocated within the investigated depth range? Which sampling design was adopted?

Lines 107-108. How was the trawl distance measured?

Lines 122-127. However, in figures S3 and S4, temperature and salinity have been reported as depth as 150 m in Bud at 63°N and Eggum at 68°N, and 250 m in Ingøy at 71°N.

Lines 130-135. It should be specified that habitat occupancy and preference refer to the sampled population between 30 and 590 m in depth. Unfortunately, the species occurrence percentage are low, mostly for G. melastomus and E. spinax (Table S1).

Line 141. Which depth strata were used to calculate COG which refers to the sampled population between 30 and 590 m in depth?

Lines 164-171. This approach should be validated by stomach content analysis since during a trawl hauls the distance covered is significant, and different species could be captured far apart from each other.

Results

Lines 175-178. In Figure S6, Eggum is 69°N while in the text is 68°N; temperature variations were assessed up to a depth of 350 m; validation at 71°N was up to a depth of 350 m.

Lines 180-181. This is somewhat strange: small variations over the years (much less than 1 degree centigrade) have proven significant, while more marked variations between shallow and deep stations (much more than 2 degree centigrade) are not.

Lines 181-182. “…a marked warming trend over time” apart from the decline around 2020.

Line 216. What class of Mollusca. Cephalopods are also Mollusca.

Line 223. M. poutassou (Blue whiting).

There are two duplicate figures (S6-S7, S8-S9) in Supporting information.

Reviewer #2: The paper entitled “Moving North: Warmer waters expand populations of deep-water cartilaginous fishes into Arctic waters.” is an interesting work and it analyses the northward geographical shift of three deep-water species across 26 years data in the Norwegian waters. The paper is well written, concise and tight to the data collected with a proper statistical framework for data interpretation. Deep water mass warming appeared as the first driver in determining such a shift, both directly (i.e. species migrate northward to accommodate their preferential thermal niche) and indirectly (by the shifting distribution of their potential prey as well). On this last aspect, I have some concerns: 1) when authors inferred the trophic driver, they based their hypothesis on the co-occurrence of prey censed in catch mass where targeted cartilaginous species were present, i.e they considered all the prey that could be entrapped by a trawl mesh size of 20 mm. To my knowledge, common prey of G. melastomus and E. spinax can be also smaller than this size as being many taxa of micro crustaceans (amphipods, mysids, euphasids, ostracods, etc), polychaetes (eunicidae, etc), cephalopods (sepiolidae, decapods, octopuses, etc), particularly in early life stages of these species. 2) the co-occurrence of larger prey (fish in particular) can be a valid proxy for the larger specimen (subadults and, the most, adults sharks) only, as ontogenetic shifts in diet have been demonstrated for G. melastomus and E. spinax, chiefly in the Mediterranean basin (prey dimension, and corresponding taxa, increases, and changes, according to the increase in fish dimension, within a generalistic-opportunistic feeding strategy that is a common trait of the species considered (the sharks in particular). Of course, the Mediterranean is a very different environment from Atlantic Ocean with a deep-water bottom temperature being much higher than oceanic corresponding depth strata (Med waters below 200 m depth display a temperature that is maintained constant at 13.5-14.5 °Celsius due to heat pump effect played by the interchange of water masses with the Atlantic; and this temperature threshold is increasing as well in the basin starting form 2000). However, ontogenetic shifts might be considered for these species in the Atlantic Ocean also such that this aspect should be carefully considered. For instance, did the authors collected information on size distribution of the samples, and/or did they perform parallel stomach content analysis on the samples investigated? For instance, comparing size-frequency distributions of different samples collected across the 26 years data set could add important info on changes in population dynamics (if you find that similar size distributions of animals are progressively found northward during the considered time span, then all life stages of the species have been moving northward and so on). Again, if you have performed parallel stomach content analysis, you can have a more complete picture of feeding habits and dietary ontogenetic shifts, i.e. a clearer idea of the prey species (not only bony fish) to consider in the catch mass if the co-occurrence is the only way available to infer the trophic factor as a driver of the northward shift of the species considered. Discussing such aspects could be an added value when claiming the trophic effect as a driver determining the geographical shift.

Some additional tips are provided in the revised pdf version of the paper

**Do you want your identity to be public for this peer review?** For information about this choice, including consent withdrawal, please see our Privacy Policy

Reviewer #1: No

Reviewer #2: **Yes:** Umberto Scacco

---

## [Author Response · Author response to Decision Letter 1]

4 Dec 2025

We would like to sincerely thank the reviewers, reviewer 1 and Dr. Umberto Scacco for their time, careful evaluation, and constructive feedback, which have greatly contributed to improving the quality and clarity of our manuscript. We have carefully considered all comments and made the corresponding revisions throughout the text. In our responses below, the line numbers refer to the revised clean version of the manuscript (without tracked changes).

Reviewer 1

Major comments

The study is of high scientific interest. It is well organized and the results are clearly presented. With regard to the methods, the experimental sampling design should be explained, indicating which statistical approach was used to allocate the stations within the investigated depth range.

The main limitation of this study is that it demonstrates the northward shift of the most superficial fraction of the population of the three species considered. In fact, although the study deals with three deep-sea species, the data sampling partially covers the depth range inhabited by these species. Therefore, some measures of distribution and abundance, such as COG, habitat occupancy and preference, relate to the sampled population in the investigated depth range. It is very likely that they don’t represent the population distributed over the whole depth gradient.

I agree that the authors included a section on the limitations of the study. The aforementioned issues regarding the bathymetric range explored also raise the question of whether populations have moved even deeper than estimated.

Answer: We thank the reviewer for this constructive comment and for highlighting these important considerations. We have now clarified the experimental sampling design in the Methods section, specifying how stations were distributed across the investigated depth range and which statistical approach was used (see line 97).

We also expanded the discussion of the study’s limitations to address the partial coverage of the depth range occupied by these deep-sea species and the potential underestimation of deeper shifts (see lines 281–283). However, we also wanted to add that in Norwegian waters, these species have been primarily observed within the temperature and depth ranges sampled in the present study (Jac et al., 2022), supporting the relevance of the investigated strata for assessing their distributional changes.

Specific comments

Line 53: “…ability of the species to exploit new resources…” in presence of predators and competitors in areas of new location.

Answer: We agree that the ability of a species to exploit new resources is strongly influenced by biotic interactions in the new environment, including competition and predation. We have therefore revised the sentence to acknowledge these ecological factors (lines 51-54) “The magnitude of these impacts depends on the sensitivity (e.g., thermal tolerance), dispersal capacity (e.g., migratory versus non-migratory), and ability of the species to exploit new resources (generalist versus specialist feeder) and habitats while interacting with resident predators and competitors in newly colonized area.”

Line 92-94: Were the data from the two different periods collected using the same sampling design, with the same tool, and within the same depth range?

Answer: Yes, data from both periods were collected using the same sampling design, gear type, and within the same depth range. We have clarified this information in the manuscript (lines 95-98).

Line 96: 800 ma?

Answer: “ma” has been removed

Line 103: A bias in the catches is also expected because the investigated depth range does not adequately cover the depth range in which the species are distributed, which is well greater than 1000 m, and 2000 m for E. spinax. Indeed, the species occurrence percentage are low, mostly for G. melastomus and E. spinax. Furthermore, sampling does not allow for the observation of any seasonal variations in the distribution of species throughout the year.

Answer: We fully agree that the investigated depth range does not encompass the entire bathymetric distribution of the studied species. We have discussed this important limitation in the Materials and Methods (Line 97) and Discussion (Lines 281-283) sections, highlighting that some bias in the catches is expected because the sampling depth range (down to 500 m) does not fully cover the depths typically inhabited by these species. However, Jac et al. (2022) showed that, in Norwegian waters, these three species are predominantly found within this depth range.

Line 104-107: The bathymetric range investigated should be also reported in the text. How were the hauls allocated within the investigated depth range? Which sampling design was adopted?

Answer: We added this information in line 97.

Lines 107-108: How was the trawl distance measured?

Answer: We have rewritten the sentence in line 110-111 to clarify that all information regarding the calculation of trawl distance and abundance is already described in Jac et al., 2022. “Abundance (individuals per km²) was standardised using trawl distance and a mean trawl opening width of 25 m, for more detail see [28].”

Line 122-127: However, in figures S3 and S4, temperature and salinity have been reported as depth as 150 m in Bud at 63°N and Eggum at 68°N, and 250 m in Ingøy at 71°N.

Answer: This point has been clarified in the revised manuscript. We now specify in lines 127-129 that, although observational data are available down to 300 m at all three sites, the corresponding GLORYS model data extend only to 150 m at Bud and Eggum and to 250 m at Ingøy, due to the model’s simplified bathymetry and relatively coarse horizontal resolution. In addition, a sentence has been added in lines 132-134 to note that the hydrographic data used in our analyses at 150, 250, and 350 m were retrieved from model grid points located slightly farther offshore in deeper areas.

Line 130-135: It should be specified that habitat occupancy and preference refer to the sampled population between 30 and 590 m in depth. Unfortunately, the species occurrence percentage are low, mostly for G. melastomus and E. spinax (Table S1).

Answer: We agree that habitat occupancy and preference metrics are specific to the sampled population within the depth range of 30–590 m and we addressed that in lines 281-283. However, in the Norwegian waters, these three species are predominantly found within this depth range (see Jac et al., 2022).

Line 141: Which depth strata were used to calculate COG which refers to the sampled population between 30 and 590 m in depth?

Answer: Yes, the COG was calculated for the sampled population within the depth range of 30–590 m. We have clarified this in the revised manuscript (Lines 147-151).

Line 164-171: This approach should be validated by stomach content analysis since during a trawl hauls the distance covered is significant, and different species could be captured far apart from each other.

Answer: We thank the reviewer for this comment. Another reviewer also raised a similar point, and we have incorporated this feedback into the revised manuscript. A broader discussion on the feeding ecology of the species has been added in lines 293–329, with a specific section addressing this issue in lines 304–313.

Line 175-178: In Figure S6, Eggum is 69°N while in the text is 68°N; temperature variations were assessed up to a depth of 350 m; validation at 71°N was up to a depth of 350 m.

Answer: We thank the reviewer for this observation. The latitude has been corrected from 68°N to 69°N in the text (Line 126) to match Figure S6. As clarified in the revised version, the model grid was not deep enough at the three validation sites (Bud, Eggum, and Ingøy). For the validation, comparisons between GLORYS and CTD observations were conducted at the same geographic positions to ensure consistency. However, for the statistical analyses, we prioritized using comparable depth levels across sites; therefore, GLORYS data were extracted from grid points located slightly farther offshore in deeper areas. This explains the apparent discrepancy between positions used for model validation and those used in the analyses.

Line 180-181: This is somewhat strange: small variations over the years (much less than 1 degree centigrade) have proven significant, while more marked variations between shallow and deep stations (much more than 2 degree centigrade) are not.

Answer: The apparent discrepancy arises from differences in the statistical significance of temporal versus vertical temperature variations. The small but consistent interannual variations shown in Figure S6 were statistically significant due to low within-year variability and long-time series, whereas the larger temperature differences between shallow and deep stations were not statistically significant because of higher spatial variability and fewer samples. Additionally, the autumn time series at 69°N shows a slightly higher p-value than the corresponding 71°N and annual series (Figures S6 and S7), which likely reflects local variability rather than an absence of trend.

Line 181-182: “…a marked warming trend over time” apart from the decline around 2020.

Answer: We have clarified in the manuscript that the warming trend in waters north of 64°N occurred over most of the study period, but a slight decline was observed around 2020. We also note that this recent variation highlights the importance of further investigation in future studies (lines 191-193).

Line 216: What class of Mollusca. Cephalopods are also Mollusca.

Answer: Changed it to Gastropoda

Line 223: M. poutassou (Blue whiting).

Answer: We are not sure that we understand the reviewer’s comment here. In line 230 we have included both the latin and the common name.

Supporting information: There are two duplicate figures (S6-S7, S8-S9) in Supporting information.

Answer: The figures are not duplicates: S6 and S8 represent the full-year dataset, while S7 and S9 correspond specifically to the September–November sampling season.

Reviewer 2 (Dr. Umberto Scacco)

Major comments

The paper entitled “Moving North: Warmer waters expand populations of deep-water cartilaginous fishes into Arctic waters.” is an interesting work and it analyses the northward geographical shift of three deep-water species across 26 years data in the Norwegian waters. The paper is well written, concise and tight to the data collected with a proper statistical framework for data interpretation. Deep water mass warming appeared as the first driver in determining such a shift, both directly (i.e. species migrate northward to accommodate their preferential thermal niche) and indirectly (by the shifting distribution of their potential prey as well). On this last aspect, I have some concerns: 1) when authors inferred the trophic driver, they based their hypothesis on the co-occurrence of prey censed in catch mass where targeted cartilaginous species were present, i.e they considered all the prey that could be entrapped by a trawl mesh size of 20 mm. To my knowledge, common prey of G. melastomus and E. spinax can be also smaller than this size as being many taxa of micro crustaceans (amphipods, mysids, euphasids, ostracods, etc), polychaetes (eunicidae, etc), cephalopods (sepiolidae, decapods, octopuses, etc), particularly in early life stages of these species. 2) the co-occurrence of larger prey (fish in particular) can be a valid proxy for the larger specimen (subadults and, the most, adults sharks) only, as ontogenetic shifts in diet have been demonstrated for G. melastomus and E. spinax, chiefly in the Mediterranean basin (prey dimension, and corresponding taxa, increases, and changes, according to the increase in fish dimension, within a generalistic-opportunistic feeding strategy that is a common trait of the species considered (the sharks in particular). Of course, the Mediterranean is a very different environment from Atlantic Ocean with a deep-water bottom temperature being much higher than oceanic corresponding depth strata (Med waters below 200 m depth display a temperature that is maintained constant at 13.5-14.5 °Celsius due to heat pump effect played by the interchange of water masses with the Atlantic; and this temperature threshold is increasing as well in the basin starting form 2000). However, ontogenetic shifts might be considered for these species in the Atlantic Ocean also such that this aspect should be carefully considered. For instance, did the authors collected information on size distribution of the samples, and/or did they perform parallel stomach content analysis on the samples investigated? For instance, comparing size-frequency distributions of different samples collected across the 26 years data set could add important info on changes in population dynamics (if you find that similar size distributions of animals are progressively found northward during the considered time span, then all life stages of the species have been moving northward and so on). Again, if you have performed parallel stomach content analysis, you can have a more complete picture of feeding habits and dietary ontogenetic shifts, i.e. a clearer idea of the prey species (not only bony fish) to consider in the catch mass if the co-occurrence is the only way available to infer the trophic factor as a driver of the northward shift of the species considered. Discussing such aspects could be an added value when claiming the trophic effect as a driver determining the geographical shift.

Answer: We thank Dr. Umberto Scacco for these insightful and constructive comments. We agree that this is an important point, particularly regarding the role of trophic dynamics and potential ontogenetic dietary shifts. As no comprehensive study has yet examined the feeding behaviour of Galeus melastomus, Etmopterus spinax, and Chimaera monstrosa in the Norwegian Sea, our approach aimed to provide a first step toward exploring potential trophic influences by examining the co-occurrence of prey species in trawl catches.

We acknowledge, however, that this method has limitations, particularly in representing smaller prey taxa (e.g., micro-crustaceans and polychaetes) and in accounting for ontogenetic shifts in diet. We have now incorporated additional discussion to address these aspects. Specifically, we discuss the size distribution of the sampled individuals and its implications for interpreting trophic drivers in line 263-269, and we expand on the potential dietary shifts during ontogeny and their influence on species’ spatial dynamics in line 294-303.

We also note in the revised manuscript that further studies, including targeted stomach content and stable isotope analyses, are needed to better understand the feeding ecology of these species in the Norwegian Sea and to strengthen inferences about trophic drivers of their northward expansion.

Specific comments

Line 72: Please consider these species are present in the Mediterranean basin also, within a much warmer environment (13.5-14.5 °C for all deep waters below 200 m depth due to heat-pump effect generated by water circulation and exchang with the Atlantic Ocean). Thermal niche of the species considered appears as wide taking into consideration the difference between the Atlantic and Mediterranean

Answer: We thank the reviewer for this insightful comment. However, we found it challenging to make a direct comparison between individuals from the Mediterranean basin and those observed in Norway, given the substantial differences in oceanographic conditions, population dynamics, and demographic structures between these regions.

Nevertheless, we revisited our manuscript and expanded our discussion with this idea in mind, in order to better highlight the potential breadth of the thermal niche of the species considered.

Line 80: Pleace Define the acronyms at first appearance

Answer: All acronyms have been defined at their first appearance in the manuscript.

Line 108: Did the authors collected information on size distribution of the samples along the time series considered? This can have great impact on results. In my opinion, considering abundance expressed as absolute density could hide the effect of fish size on the shifting pattern towards the no

---

## [Decision Letter · Decision Letter 1]

23 Dec 2025

Dear Dr. Jac,

Thank you for submitting your manuscript to PLOS ONE. After careful consideration, we feel that it has merit but does not fully meet PLOS ONE’s publication criteria as it currently stands. Therefore, we invite you to submit a revised version of the manuscript that addresses the points raised during the review process.

Authors should follow the reviewers suggestions in order to resolve the minor issues and make the manuscript suitable for publication.

We look forward to receiving your revised manuscript.

Kind regards,

Claudio D'Iglio, Ph.D.

Academic Editor

PLOS One

Journal Requirements:

Reviewers' comments:

Reviewer's Responses to Questions

**Comments to the Author**

Reviewer #1: (No Response)

Reviewer #2: All comments have been addressed

2. Is the manuscript technically sound, and do the data support the conclusions?

Reviewer #1: (No Response)

Reviewer #2: Yes

3. Has the statistical analysis been performed appropriately and rigorously?

Reviewer #1: (No Response)

Reviewer #2: Yes

4. Have the authors made all data underlying the findings in their manuscript fully available?

Reviewer #1: (No Response)

Reviewer #2: Yes

5. Is the manuscript presented in an intelligible fashion and written in standard English?

Reviewer #1: (No Response)

Reviewer #2: Yes

Reviewer #1: The work has been substantially improved, although some limitations, adequately addressed by the authors, remain.

In particular, the sampling design adopted is still unclear. It was not reported in line 97. What approach (random-stratified, systematic, other) was used to allocate the stations along the whole bathymetric gradient investigated? This is important for calculating the COG. In fact, if the stations have been allocated unbalanced along the whole depth gradient, the COG value may be unreliable.

Reviewer #2: I think the revised version of the paper has addressed almost all my concerns arose in the first review round. However, I also noticed that the introduction should need a brief paragraph contextualizing and stressing the importance of trophic drivers, which is later considered one of the two main factors determining northward shift of species investigated. I think this paragraph should contain general information on what is already known on species trophic habit of the investigated species at the regional but also at larger scale than regional where these species are present as well. Maybe, this could be done after line 54 of the revised introduction, but authors can place such a part wherever convenient throughout the text.

**Do you want your identity to be public for this peer review?** For information about this choice, including consent withdrawal, please see our Privacy Policy

Reviewer #1: No

Reviewer #2: **Yes:** Umberto Scacco

---

## [Author Response · Author response to Decision Letter 2]

30 Jan 2026

We sincerely thank both reviewers, Reviewer 1 and Dr. Umberto Scacco, for their time, careful evaluation, and constructive feedback, which have greatly contributed to improving the clarity and quality of our manuscript. We have carefully addressed all comments and made corresponding revisions throughout the text. Line numbers in our responses refer to the revised clean version of the manuscript (without tracked changes).

Reviewer 1

Comments

The work has been substantially improved, although some limitations, adequately addressed by the authors, remain.

In particular, the sampling design adopted is still unclear. It was not reported in line 97. What approach (random-stratified, systematic, other) was used to allocate the stations along the whole bathymetric gradient investigated? This is important for calculating the COG. In fact, if the stations have been allocated unbalanced along the whole depth gradient, the COG value may be unreliable.

Answer: We thank the reviewer for highlighting the need to clarify our sampling design. In the revised manuscript, we have specified that the stations were allocated using a systematic stratified design along the entire bathymetric gradient, covering shallow, intermediate, and deep strata (line 101-103). We have also clarified in line 174-176 that this approach ensures a balanced representation of depths across the study area, minimizing potential bias in the calculation of the center of gravity (COG).

Reviewer 2 (Dr. Umberto Scacco)

Comments

I think the revised version of the paper has addressed almost all my concerns arose in the first review round. However, I also noticed that the introduction should need a brief paragraph contextualizing and stressing the importance of trophic drivers, which is later considered one of the two main factors determining northward shift of species investigated. I think this paragraph should contain general information on what is already known on species trophic habit of the investigated species at the regional but also at larger scale than regional where these species are present as well. Maybe, this could be done after line 54 of the revised introduction, but authors can place such a part wherever convenient throughout the text.

Answer: We thank Dr. Umberto Scacco for this suggestion. In response, we have added a paragraph in the Introduction (line75-81) to contextualize the role of trophic interactions in shaping species’ distributions and northward shifts. Specifically, we now describe the diets of Chimaera monstrosa, Etmopterus spinax, and Galeus melastomus at both regional and broader Northeast Atlantic scales, highlighting that prey availability may facilitate or constrain northward movements. This addition emphasizes trophic drivers as a key factor alongside environmental variables in determining species’ range shifts. It can be read as follow “Trophic interactions further shape their distribution and range shifts: C. monstrosa feeds primarily on benthic crustaceans, molluscs, and small demersal fishes [29,30], E. spinax consumes mesopelagic fishes, cephalopods, and crustaceans [31–33], and G. melastomus feeds opportunistically on small fishes, cephalopods, and crustaceans [32,33]. These consistent dietary patterns across the Northeast Atlantic suggest that prey availability at regional and broader scales may facilitate or constrain northward movements.”

---

## [Decision Letter · Decision Letter 2]

11 Feb 2026

Moving north: Warmer waters expand populations of deep-water cartilaginous fishes into Arctic waters.

PONE-D-25-46321R2

Dear Dr. Jac,

We’re pleased to inform you that your manuscript has been judged scientifically suitable for publication and will be formally accepted for publication once it meets all outstanding technical requirements.

Kind regards,

Claudio D'Iglio, Ph.D.

Academic Editor

PLOS One

Additional Editor Comments (optional):

Reviewers' comments:

Reviewer's Responses to Questions

**Comments to the Author**

Reviewer #1: All comments have been addressed

Reviewer #2: (No Response)

2. Is the manuscript technically sound, and do the data support the conclusions?

Reviewer #1: Yes

Reviewer #2: Yes

3. Has the statistical analysis been performed appropriately and rigorously?

Reviewer #1: Yes

Reviewer #2: Yes

4. Have the authors made all data underlying the findings in their manuscript fully available?

Reviewer #1: Yes

Reviewer #2: Yes

5. Is the manuscript presented in an intelligible fashion and written in standard English?

Reviewer #1: Yes

Reviewer #2: Yes

Reviewer #1: (No Response)

Reviewer #2: The paper addressed all my concerns, in particular the part on trohic driver as factor contributing the north expansion of deep water species investigated.- I think the work is now ready for publication in PLOS One

**Do you want your identity to be public for this peer review?** For information about this choice, including consent withdrawal, please see our Privacy Policy

Reviewer #1: No

Reviewer #2: **Yes:** Umberto Scacco

---

## [Editor Report · Acceptance letter]

PONE-D-25-46321R2

PLOS One

Dear Dr. Jac,

I'm pleased to inform you that your manuscript has been deemed suitable for publication in PLOS One. Congratulations! Your manuscript is now being handed over to our production team.

Kind regards,

on behalf of

Dr. Claudio D'Iglio

Academic Editor

PLOS One